# New Computerized Method in Measuring the Sagittal Bowing of Femur from Plain Radiograph—A Validation Study

**DOI:** 10.3390/jcm8101598

**Published:** 2019-10-03

**Authors:** Chen-Kun Liaw, Yen-Po Chen, Tai-Yin Wu, Chiou-Shann Fuh, Ruey-Feng Chang

**Affiliations:** 1Department of Orthopedics, School of Medicine, College of Medicine, Taipei Medical University, Taipei City 11031, Taiwan; chenkunliaw@gmail.com; 2Division of Orthopedics, Shuang Ho Hospital, Taipei Medical University, New Taipei City 23561, Taiwan; 3Department of Orthopedics, National Taiwan University Hospital, Taipei City 10002, Taiwan; 4Anesthesia Department, Taipei Hospital, Ministry of Health and Welfare, New Taipei City 24213, Taiwan; apochen@gmail.com; 5Department of Computer Science and Information Engineering, National Taiwan University, Taipei City 10617, Taiwan; fuh@csie.ntu.edu.tw; 6Department of Family Medicine, Renai Branch, Taipei City Hospital, Taipei City 10629, Taiwan; dienewu@yahoo.com.tw; 7Institute of Epidemiology and Preventive Medicine, National Taiwan University, Taipei City 10055, Taiwan

**Keywords:** femur bowing, region growing territory method, bowing mismatch

## Abstract

Background: Mismatch of intramedullary nails with the bowing of femur is a frequent clinical finding. Previous studies showed inconsistent results. Methods: We present an algorithm of region growing territory method to get the radii of the anterior bowing of femur. We also tested it on ten radiographs. Plain radiographs of the lateral view of femur from five men and five women taken between January and August 2014 in Taipei Hospital were chosen randomly. The curvature of femur outline and medullary canal were measured for three times each. Radii of curvature of whole femur, proximal, middle and distal parts were calculated and analyzed. Results: The coefficient of variation of the 240 measurements ranged from 0.007 to 0.295 and averaged 0.088. The average radii of curvature of the whole, proximal, middle, and distal femur were 1318 mm, 752 mm, 1379 mm, and 599 mm, respectively. At the distal part of the femur, the radius of curvature of the femur outline (452 mm) was smaller than the medullary canal (746 mm) (*p* < 0.05). Women’s femur was straighter than men’s when we compared the whole length (1435 mm vs. 1201 mm, *p* < 0.05). The radii we calculated were smaller than the current intramedullary nails. Conclusion: The results showed that the inter-observer and intra-observer differences are acceptable, support the impression that different bowing conditions existed for Asians as compared to Caucasians, and also indicate the mismatch of current instruments to the curvature of femur.

## 1. Introduction

The difference of sagittal bowing of femur among different individuals or even different ethnicities is important in present orthopedic practice. Orthopedic surgeons often use the same implants when they are operating femoral fractures without considering the differences of sagittal bowing. However, mismatch of the intramedullary nails or rods with the anterior bow of the femur causes problems during the surgery and throughout the healing and rehabilitation period.

Various studies aimed at calculating the curvature of the femoral bones. However, most of the studies were not about Asian people. The results were also quite diverse. Egol et al. reported an average radius of curvature of 1200 mm ± 360 mm in New York, USA [1]. In this study, African Americans had a greater average radius of curvature than Caucasians, and men had a greater average radius of curvature than women (African American men (1320 mm), African American women (1330 mm), Caucasian men (1190 mm), and Caucasian women (1050 mm), respectively). Harma et al. reported that the average Caucasian medullary femoral bowing was 722.37 mm ± 230 mm in Anatolia, Turkey [2]. Su et al. measured 426 Chinese femurs by three-dimensional analysis and reported that the average radius of femoral curvature was 971.44 mm ± 211.68 mm in Beijing, China [3].

Tang et al. segmented the femur into three parts (proximal, middle, and distal) and calculated the three radii [4]. They worked on 100 radiographs of Chinese patients and concluded that the lateral contour of the Chinese femur is like a hockey stick. The average radii of curvature of the proximal, middle, and distal one-third of the femur were 1081.6 mm, 926.2 mm, and 715.1 mm, respectively. However, the average radius of the whole femur was not reported [4]. There were some other data from different studies. The average radius of 1144 mm was reported from Harper and Carson [5], 1090mm was found by Johnson et al. [6], and 1410 mm was suggested from Bruns et al. [7], respectively.

In addition to the variated results, the measuring methodology was also different. Regarding the studies which did the measurements on the plain radiographs, both Egol et al. [1] and Harma et al. [2] were just taking three points on the radiograph of the femur which were at the level of the lesser trochanter, flare of the condyles, and the midpoint of the two points, respectively. After the three points were marked, they drew a circle through these three points to get the radius. Tang et al. used three computer programs to do their measurement [4]. First, they used the software CorelDRAW to determine the outer cortex of the femur. Second, they used a stand-alone skeletonization program which was developed by Costa et al. [8] to obtain the midline of the femur. This midline served as the skeleton curve. Third, they used the program Unigraphics version 17 to define 60 sample points on the skeletal curve. These 60 points were divided into three equal parts as 20 points each. Finally, the radius of curvature of each part was determined from the 20 points by curve analysis function.

Buford et al. reported a method using computer tomography, built a three-dimensional model, then selected three points to calculate the curvature [9]. Park et al. used computer tomography and printed a real three-dimensional model for testing [10]. Thiese et al. also used computer tomography, built the model, and then calculated the radius [11]. These computer tomography methods did not separate the curvature to three parts as Tang et al. did [4].

In this study, we propose a new method to measure sagittal bowing of femur. With computer assistance, we developed a new algorithm to determine the curvature of the femur. To aid clinical practice, we aimed to measure the curvature of femur precisely.

## 2. Experimental Section

We retrieved plain radiographs of the lateral view of femur taken at Taipei Hospital, Ministry of Health and Welfare, Taiwan from January to August 2014. Initially, we retrieved 93 cases from the Image bank. The radiographs of participants with the age of more than 75 (*n* = 9) or less than 18 (*n* = 1) years old were excluded because of the special regulations of local institutional review board (IRB). Nine cases with fractures of the femur with notable displacement were excluded, too. On examining the images, the whole length of the femur had to be included in the image, the outer border of the femur and the border of the medullary canal had to also be easily outlined. Otherwise these images were excluded as well (Figure 1). Finally, 35 radiographs met the inclusion criteria (Figure 2).

Of the included 35 radiograms, we randomly chose images of five men and five women to be included in the study. The selected ten images were saved as DICOM (Digital Imaging and Communications in Medicine) format without compression. The resolution was 0.15 mm per pixel. With Windows operating system (Window 10, Microsoft, Redmond, WA, US) and Delphi 2010 language, we built a software program to read the image which automatically calculated the radii of curvatures of the femur as output after we drew the border of the femur outline and the medullary canal.

After the image was read into the program, we determined the proximal and distal ends of the femoral shaft outline and the medullary canal. Then we drew the upper and the lower border of both the femoral shaft outline and the medullary canal between the proximal end and the distal end. Because the femur is in oblique position in the lateral view of the radiography, we regarded the anterior and posterior sides as the upper side and lower sides, accordingly. For the femoral shaft outline, the distal ends of the upper and lower borders were defined as the junctions of the femoral shaft and the condylar region on both sides (Figure 3 blue points 1 and 2). After we got the two distal ends, we drew a line between them. The junction of this line and both borders of the medullary canal were defined as the distal end of the border of the medullary canal (Figure 3 yellow points a and b).

The proximal end of the lower border of the femoral shaft outline was defined as the bottom of the lesser trochanter (Figure 3 blue point 3). After we got this point, we drew a line through this point and measured both angles of this line and both borders of the femoral shaft outline. When these two angles were equal, the point of intersection of this line to the upper border of the femoral shaft outline was defined as the proximal end of the upper border of the femoral shaft (Figure 3 purple point 4). The intersections of this line to both the upper and the lower borders of the medullary canal were defined as the proximal ends of both borders of the medullary canal (Figure 3 yellow points c and d).

Once we had determined the proximal and distal ends, we started to draw the upper and lower borders on the image from the proximal end to the distal end, respectively. For each image, we drew the outlines of the femur and the borders of the medullary canal to calculate the curvature of the femur (cortical bow) (Figure 4a) and that of the interior medullary canal (medullary bow) (Figure 4b).

We introduced an algorithm of region growing territory method to find the center of the femur outline and the interior medullary canal. The region growing algorithm has been widely used in image segmentation [12,13,14,15]. The central line of the two curves is similar to segmentation. Thus, we modified it to region growing territory method. We set two curves as two seeds. The seeds grow and then meet in the central line. The region growing territory method is described below:

After we drew the upper and the lower borders of the femur outline, the predefined area then started to enlarge the occupied territory from both sides (borders) spontaneously with the same speed. When the two territories met each other, we took this point as the center of the upper and the lower border. By repeating this process, we got a line which consists of the center points of upper and lower borders (Figure 5 pink line). Finally, we calculated the curvature of this central line and got the radius which is the curvature of the femur. We also divided this central line into 3 equal segments: proximal, middle, and distal parts and calculated the radii of each segment separately to get the radii of each part. We then have data of four radii for each measurement. Similarly, we were able to get data of the radii of curvature of the medullary canal. These measurements and calculations were all integrated into the computer program. The results were displayed on the screen as a scheme (Figure 6).

Each measurement was repeated three times by one of the author (C.K.L.) twice with one week interval and then by the other author (Y.P.C.). We calculated the coefficient of variation of each three measurement. Smaller coefficient of variation means good intra-observer and inter-observer reliability.

Ethics: This study was approved by the Institutional Review Board of Taipei Hospital (TH-IRB-0015-0001).

## 3. Results

Of the 10 radiographs, there were 5 men and 5 women. The average age was 34 years old (SD = 10.9 years, range 20 to 57). We did the measurement 3 times for the femur outline and 3 times for the medullary canal of each radiograph. There were 60 measurements in total. We got 4 radii from each measurement: whole, proximal, middle, and distal parts. The coefficient of variation of these 240 measurements ranged from 0.007 to 0.295 and averaged 0.088. Our analysis showed that the three different measurements on the same radiograph were all highly correlated. The coefficient of variation of 0.088 was rather good for our purpose because the magnification of radiographs was 1.10× to 1.20×, larger than our results.

If we looked at the data of femur outline and medullary canal together, the average radii of curvature of the whole, proximal, middle, and distal were 1318mm (range 872 mm to 1979 mm, SD 297 mm), 752 mm (range 362 mm to 1305 mm, SD 212 mm), 1379 mm (range 873 mm to 1964 mm, SD 288 mm), and 599 mm (range 289 mm to 1132 mm, SD 220 mm), respectively. The distal one-third of the femur was the most bowed portion, followed by the proximal one-third portion. The middle-third portion was the straightest. If we separated the femur outline and medullary canal into two groups, the distal one-third was still the most bowed portion in the femur outline group, while the proximal one-third became the most bowed portion in the medullary canal group. The middle-one third portion was also the straightest in two groups. There was no difference of the radii of the femur outline and the medullary canal in whole length, proximal and middle parts (*p* > 0.05) except for the distal part (*p* < 0.05). The radius of the distal part of femur outline was significantly different from that of medullary canal (452 mm, SD 133 mm vs. 746 mm, SD 190 mm).

In addition, we compared radii between both genders and observed that the whole length of women’s femur was straighter than men’s (1435 mm vs. 1201 mm, *p* = 0.002). However, there was no significant difference in the proximal, middle, and distal parts between men and women (*p* = 0.283, 0.456, and 0.135, respectively). The data are shown in Table 1 and Table 2.

## 4. Discussion

We were interested in measuring the sagittal bowing of femur because the mismatches of the implants and the femur caused technical difficulties encountered during the femoral surgeries or complications [16,17]. Angular defects [6,18], iatrogenic fractures [19,20], and penetration of the distal anterior femoral cortical bone [18,21] were the major complications. Some studies measured sagittal bowing of the femur [1,2,5,6,7]. However, these studies were mostly done in Western countries. Studies about measurement of the femur of Asian or Chinese people are relatively scarce. In addition, there has been no study investigating the difference of the sagittal bowing in different segments of the femur before Tang. Tang et al. first introduced the idea to divide the femur into proximal, middle, and distal parts and measured the 3 radii separately [4]. There were different conditions relating to the mismatches. Hwang et al. reported 4 cases with mismatch between proximal femoral nails (PFNs) and medullary canals which resulted in medial wall fracture when inserting the PFNs [22]. Egol et al. concluded that the curvatures of current intramedullary nails were mismatched to average femoral curvature [1]. In other conditions such as total knee arthroplasty (TKA), sagittal bowing of the femur also affected the outcome of the surgery [4,23].

PFN is mainly placed in the proximal part of the femur. The intramedullary nail is inserted from the proximal end of femur and passes through the middle part of the femur. The femoral component of the implant for TKA is inserted and fixed at the distal part of the femur. Thus it is very important to distinguish and to consider the radii of the three different parts of the femur in clinical practice.

We did the measurements for the radii of the whole femur and the three segmentations: proximal, middle, and distal. The radius of the whole femur is similar to the result of Egol et al. (1318 mm ± 297mm vs. 1200 mm ± 360 mm) and is straighter than Harma et al.’s report (only 722.37 mm ± 230 mm). Our result is also straighter than the result of Su et al. (971.44 mm ± 211.68  mm).

There was no report of the curvature of the medullary canal in Tang et al.’s stud. Both Tang’s and our study divided the femur into 3 segments and measured the 3 values. The results were different. In Tang’s study, it is most bowed in the distal one-third of the femur, followed by the middle one-third and then the upper (proximal) one-third. The values were 715.1 mm ± 77.0 mm, 926.2 ± 117.4 mm, and 1081.6 mm ± 225.3 mm, respectively. In our measurements, the distal third was the most bowed part as well (599 mm ± 220 mm). The second bowed part was the proximal one-third (752 mm ± 212 mm), and the middle one third was the straightest part (1379 mm ± 288 mm).

Was the distal one-third of the femur the most bowed part of the femur? Yehyawi et al. provided another result [23]. They measured the angles of curvature of the proximal, middle, and distal thirds of the femur and got the values of 5.2°, 0.3°, and 3.1°, respectively. In their study, the proximal third was the most bowed part, followed by the distal one third, and the straightest part was the middle one-third. Because Yehyawi et al. calculated the angle of curvature and did not mention about the radii of the parts of the femur, we were not able to compare the radii. However, the result of our study supports Tang et al.’s suggestion that the distal segment of the femur of the Caucasians did not have distal sagittal bowing sufficiently conspicuous for surgeons to recognize while in Chinese patients, distal sagittal bowing is a constant and important feature. It also implies that the distal femur in Asian people, both Chinese and Taiwanese, may be more bowed than in Western people [24].

It is surprising that the proximal one-third part of the femur was the straightest part of the radius (1081.6 mm ± 225.3 mm) in Tang’s study, while this value was only (752 mm ± 212 mm) in our study [4]. Tang’s study was done in Hong Kong, and their samples were all recognized of Chinese origin [4]. Our study was done in Taiwan, and the samples were all Taiwanese who are generally considered as Chinese, too. We have no idea whether the difference was caused by different measuring methods or whether the radius of the proximal one-third of the femur was really different between Hong Kong Chinese and Taiwanese Chinese. In both measurements, even though the calculation methods were different, the definitions of proximal and distal ends were the same. However, there is one thing worthy of further research. The cases in Tang’s study all suffered from knee joint arthritis: 85 osteoarthritic knee arthritis and 15 rheumatoid knee arthritis, while the cases in our study were taken randomly from the general population without specific systemic or chronic disease. Since knee arthritis is a chronic disease and patients usually suffered for a very long time before surgery, long-term knee arthritis may reasonably affect sagittal bowing of the femur.

The radii of curvature of current intramedullary nails range from 1860 mm to 2950 mm [1]. In comparison with the radius of anterior bowing, we measured which were only 1318 mm for the whole length and just 752 mm for the proximal part; there is indeed a significant mismatch. For revision prostheses for total knee arthroplasty in current use, the radius of curvature ranged from 1300 mm to 2000 mm [1], but the average radius of the distal part of the femur we measured was only 599 mm. There was also significant mismatch between the current knee implants and the radii of the femur, too. As we measured the femur outline and medullary canal separately, we noticed that there is no difference between the curvatures of whole length, proximal, and middle parts of the femur outline and medullary canal (*p* > 0.05). We might take the average radius value for use when we are inserting the intramedullary nails, but these two radii of curvature were significantly different at the distal part (452 mm vs. 746 mm, *p* < 0.005). Therefore, we suggest taking the radius of curvature of medullary canal instead of the average value into consideration when we are doing total knee arthroplasty.

Finally, since we did the measurements of the same body part on the same radiograph for three times and the results were highly correlated, we concluded that our method is a stable and precise way to measure the curvature of the femur. Furthermore, in comparison with other studies such as measurements of Egol et al. [1] and Harma et al. [2] which were just taking three points from a femur photography, or that of Tang et al. which took only 20 points of each part of the femur to calculate the radius of curvature, our method could approximate the real curvature of the femur more than any other study [4].

This study was designed to provide reliability, so the patient number is small. In the future, we will enlarge our sample size and try to establish a bigger database building up the basic profile of the sagittal bowing of femur in Taiwan. We may also compare with results of other Asian countries like Japan, Korea, or the Philippines to see if there are significant differences of the femur among these Asian countries. After the database is established, we might cooperate with the manufacturers of orthopedic implants to develop specialized types of various sizes and reduce the mismatch between the instruments and the femur. Finally, we might be able to accomplish the goal of having a better outcome of the related surgery and help the patients heal and rehabilitate better.

## 5. Conclusions

We presented a new computer method to determine curvature of femur. The results showed good reliability. We plan to do a larger scale study in the future to reduce mismatch of implants.

## Figures and Tables

**Figure 1 jcm-08-01598-f001:**
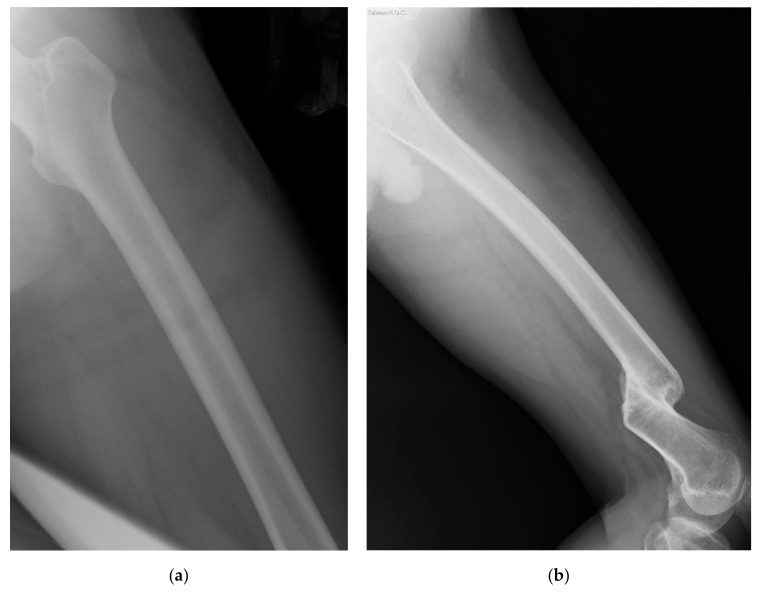
(**a**) Two examples of the excluded images: Note the whole length of femur was not included in the image; (**b**) Fracture of the femur with notable displacement is noted in the image.

**Figure 2 jcm-08-01598-f002:**
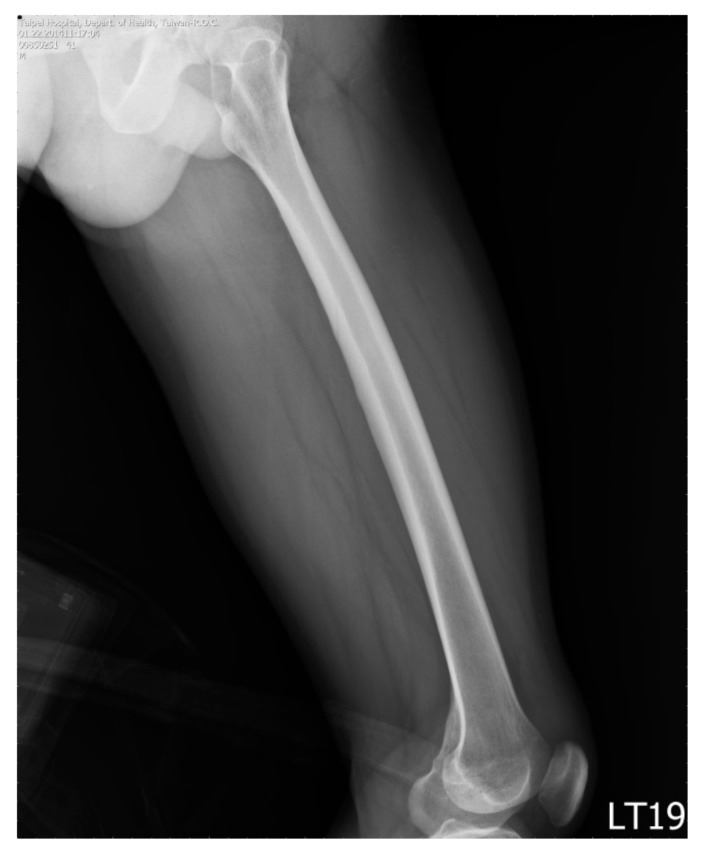
An example of the included images.

**Figure 3 jcm-08-01598-f003:**
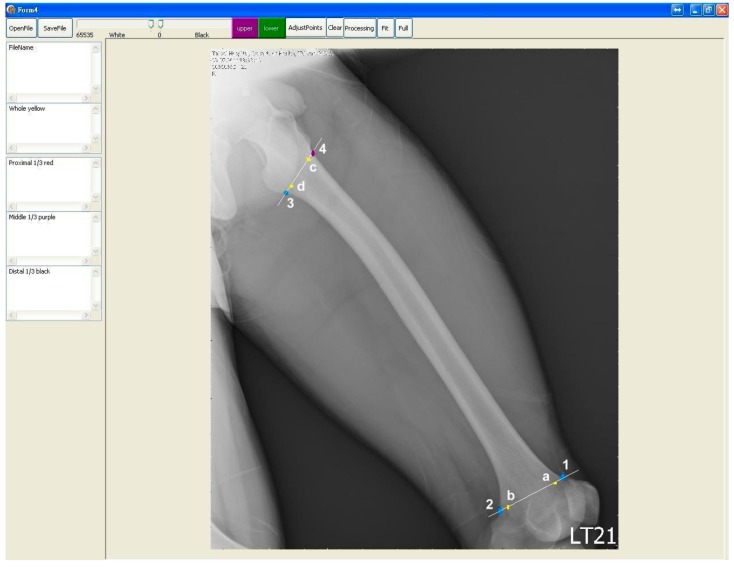
Blue points 1 and 2 were marked as the distal ends of cortical border (femoral shaft outline). Blue point 3 was marked below the lesser trochanter. Purple point 4 was determined on the other side. Both points 3 and 4 were the proximal ends of cortical border. The 4 yellow points were the distal and proximal ends (a, b and c, d) of borders of the medullary canal.

**Figure 4 jcm-08-01598-f004:**
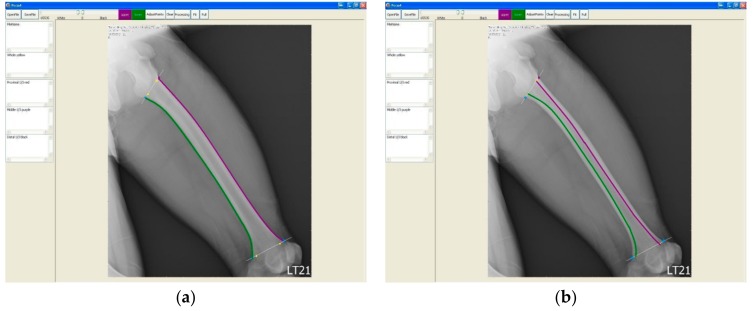
The marked (drawn) outline of cortical border (**a**) and the marked (drawn) outline of medullary canal (**b**).

**Figure 5 jcm-08-01598-f005:**
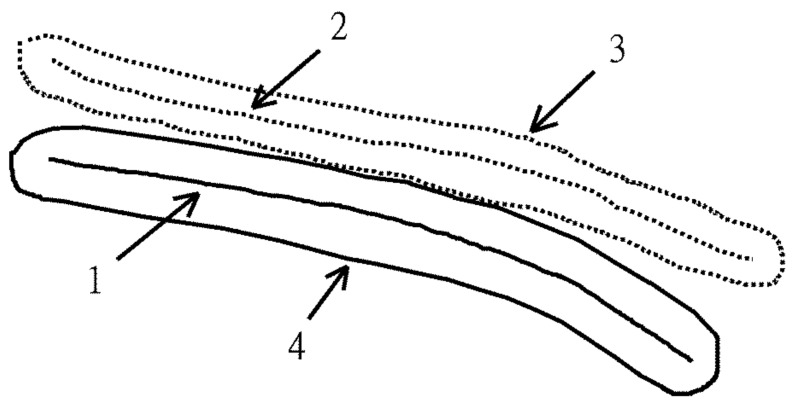
Region growing territory method. Solid line 1 represents one curve. Dot line 2 represents another curve. 1 and 2 are seen as seed and grow to two territories (3 and 4). The two territories meet and form central curve. If they do not meet, they will keep on growing.

**Figure 6 jcm-08-01598-f006:**
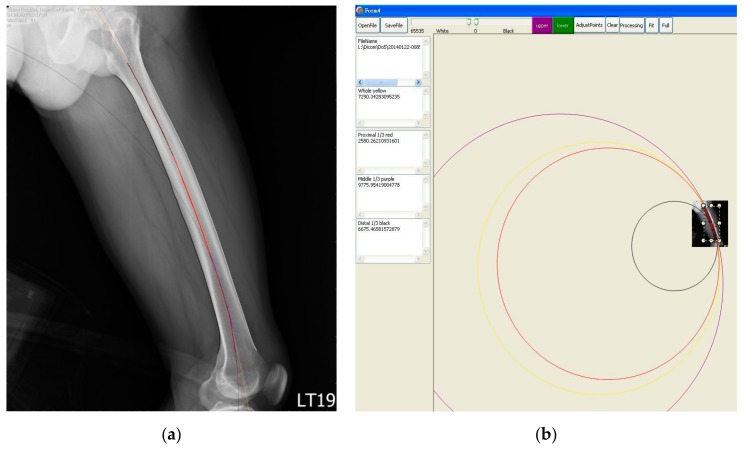
Result screen: zoom in (**a**), zoom out (**b**).

**Table 1 jcm-08-01598-t001:** General data (F: female. M: male). Units were mm.

Age	Gender	Whole Outer	Whole Inner	Proximal Outer	Proximal Inner	Middle Outer	Middle Inner	Distal Outer	Distal Inner
24	F	1631	1883	340	811	1883	1404	653	671
27	F	1338	1600	379	1043	1546	1577	983	712
38	F	1105	982	311	632	1089	967	673	452
20	F	1730	1604	432	574	1671	1520	1041	693
31	F	1299	1183	770	1005	1193	1217	604	629
Female’s mean		1421	1450	446	813	1476	1337	791	631
Female’s SD		255	362	186	212	331	248	204	105
37	M	1696	1614	392	632	1486	1812	907	871
40	M	1087	899	436	696	1179	1172	746	1032
43	M	1094	1117	379	795	1418	1166	1098	905
57	M	1112	1038	534	758	1068	1117	715	693
27	M	1330	1019	550	510	1413	1676	558	405
Male’s mean		1264	1137	458	678	1313	1389	805	781
Male’s SD		262	278	80	113	180	329	206	243

**Table 2 jcm-08-01598-t002:** Radii of different parts in different groups.

Measurement of Femur	Total (*n* = 60)	Femur Outline (*n* = 30)	Medullary Canal (*n* = 30)	Men (*n* = 30)	Women (*n* = 30)	*p*-value
Whole	1318 ± 297	1342 ± 252	1294 ± 338	1201 ± 257	1435 ± 291	0.002
Proximal	752 ± 212	798 ± 210	706 ± 208	793 ± 227	711 ± 192	0.283
Middle	1379 ± 288	1394 ± 285	1363 ± 294	1351 ± 277	1407 ± 300	0.456
Distal	599 ± 220	452 ± 133	746 ± 190	568 ± 161	630 ± 266	0.135

Unit: mm, expressed as mean ± SD.

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
