# Peer review of "New Computerized Method in Measuring the Sagittal Bowing of Femur from Plain Radiograph—A Validation Study"

_jcm, 2019, doi:10.3390/jcm8101598_

Round 1

Reviewer 1 Report

General comments:

In this study, Liaw et al develop a software-based program for quantification of sagittal femoral curvature, then test this software in the measurement of 5 male and 5 female radiographs.  While the results are interesting and add to the current literature on this topic, several major issues remain to be addressed.  For example, key literature that could better justify this study is overlooked in the introduction section. Males and females are currently pooled for analysis, but should be separated so that sex-specific differences can be appreciated.  It is unclear how or when the authors will make this software available to other researchers, limiting interest in its clinical utility.  Finally, there are numerous grammatical and typographical errors that made this document difficult to review; it is necessary that the authors seek professional, English-language editing prior to resubmitting this manuscript.

Specific comments:

Introduction: the authors indicate in the introduction that it’s important to know sagittal curvature of the femur as it can affect the efficacy of intramedullary nails. However, they fail to cite two key Pubmed-indexed articles which highlight this relationship between femoral curvature and intramedullary nailing: Buford et al J Orthop Trauma 2014 (PMID: 24121983) and Park et al Injury 2016 (PMID 26980645).  The authors should read these papers and add a paragraph to the introduction summarizing their purpose and findings, as they better justify the authors’ efforts in the current manuscript.
Experimental Section, line 83: The authors exclude any radiographs from individuals over age 75 or under age 18, but provide no justification for why these age ranges were excluded. Please provide justification for the ages selected for study, and those selected for exclusion. Also, according to the 1st Figure 1, it appears that the authors excluded femurs which demonstrated a fracture that affected curvature. This exclusion criteria is not currently stated, but should be.
Experimental Section, line 87: The authors indicate that “demographic data were recorded,” but to this reviewer’s knowledge, are not presented anywhere except as the mean +/- SD of age. The authors should instead provide a table listing demographic information of the males and females selected for study, as well as including the demographic data for all 35 available radiographs.  Also, demographic data should be separated between males and females, as should the results.
Figure 1: Two separate Figures labeled Figure 1 are presented, which is confusing. These images should be combined into one figure and presented as such.
Experimental Section, line 93: The authors randomly chose 5 male and 5 female radiographs of the 35 available for their study. It is not explained why only 10 of the 35 were studied.  Did the authors have sufficient power with n=5 per sex? This should be addressed. In addition, it should be justified why only 10 of the 35 available samples were selected for study. How does this sample size compare to other previously published studies of femoral curvature?
Experimental Section, line 96: the authors indicate that they built a software program, which is subsequently described in the following paragraphs. However, it is not indicated HOW this program was built (e.g., programming language, operating system it is compatible with, etc.), nor whether the program has been / will be deposited into a publically available database for use by others. These issues must be noted and addressed.
Figure 2: Actually, I am assuming this is Figure 2, as its current label is “Fig Error! No text or specified style in document.” This must be corrected.
Experimental Methods, line 128: the “algorithm of territory method” needs to be more fully described, diagramed, and referenced. It is unclear if the authors created this algorithm, or borrowed it from a previously published study.  
Figure 5: the “pink line” is difficult to discern. It is recommended that the authors change this pink line to something “dashed” or otherwise, so it is easier to see when printed in black and white (as many readers may do).
Results, line 154: The authors indicate that all measurements from the same radiograph were highly correlated but do not present data to support this statement. Please provide.
Results, general: At present, the findings from the authors’ study are limited.  It may increase the merit of the study to break the results into male and female, then statistically compare the femoral curvature of males vs. females. This would add rigor and interest to the study, rather than simply demonstrating a new tool for measuring femoral curvature, because at present it is unclear from the present results whether this new tool for measuring femoral curvature represents an advance or better alternative to currently available methods as previously publisihed.

Author Response

Response to Reviewer 11 Comments

Point 1: Introduction: the authors indicate in the introduction that it’s important to know sagittal curvature of the femur as it can affect the efficacy of intramedullary nails. However, they fail to cite two key Pubmed-indexed articles which highlight this relationship between femoral curvature and intramedullary nailing: Buford et al J Orthop Trauma 2014 (PMID: 24121983) and Park et al Injury 2016 (PMID 26980645).  The authors should read these papers and add a paragraph to the introduction summarizing their purpose and findings, as they better justify the authors’ efforts in the current manuscript.

Response 1: We read these manuscripts and included in the introduction section. Thank you for reminding us.

Text change:

(Introduction section line 79 to 84)

Buford et al. reported a method using computer tomography, built a three-dimensional model, then selected three points to calculate the curvature [2]. Park et al. used computer tomography and printed a real three-dimensional model for testing [15]. Thiese et al. also used computer tomography, built the model and then calculated the radius [19]. These computer tomography methods did not separate the curvature to three parts as Tang et al. did [18].

Point 2: Experimental Section, line 83: The authors exclude any radiographs from individuals over age 75 or under age 18, but provide no justification for why these age ranges were excluded. Please provide justification for the ages selected for study, and those selected for exclusion. Also, according to the 1st Figure 1, it appears that the authors excluded femurs which demonstrated a fracture that affected curvature. This exclusion criterion is not currently stated, but should be.

Response 2: Our method is designed to measure radiographs with clear bone cortex edge. However our IRB has special regulations for those over age 75 or under age 18 and requires their patron’s consent. To make our study easier, we choose to avoid these regulations.

Text change:

(Experimental Section line 89 to 93)

We retrieved plain radiographs of the lateral view of femur taken at Taipei Hospital, Ministry of Health and Welfare, Taiwan from January to August 2014. Initially, we retrieved 93 cases from the Image bank. The radiographs of participants with the age of more than 75 (n=9) or less than 18 (n=1) years old were excluded because of the special regulations of local institutional review board (IRB).

Point 3: Experimental Section, line 87: The authors indicate that “demographic data were recorded,” but to this reviewer’s knowledge, are not presented anywhere except as the mean +/- SD of age. The authors should instead provide a table listing demographic information of the males and females selected for study, as well as including the demographic data for all 35 available radiographs. Also, demographic data should be separated between males and females, as should the results.

Response 3: We added a table for general data of the ten radiographs.

Text change: (Table 1)

Table 1. General data (F: female. M: male). Units were mm.

Age

Gender

Whole Outer

Whole Inner

Proximal Outer

Proximal Inner

Middle Outer

Middle Inner

Distal Outer

Distal Inner

24

F

1631

1883

340

811

1883

1404

653

671

27

F

1338

1600

379

1043

1546

1577

983

712

38

F

1105

982

311

632

1089

967

673

452

20

F

1730

1604

432

574

1671

1520

1041

693

31

F

1299

1183

770

1005

1193

1217

604

629

Female’s mean

1421

1450

446

813

1476

1337

791

631

Female’s SD

255

362

186

212

331

248

204

105

37

M

1696

1614

392

632

1486

1812

907

871

40

M

1087

899

436

696

1179

1172

746

1032

43

M

1094

1117

379

795

1418

1166

1098

905

57

M

1112

1038

534

758

1068

1117

715

693

27

M

1330

1019

550

510

1413

1676

558

405

Male’s mean

1264

1137

458

678

1313

1389

805

781

Male’s SD

262

278

80

113

180

329

206

243

Point 4: Figure 1: Two separate Figures labeled Figure 1 are presented, which is confusing. These images should be combined into one figure and presented as such.

Response 4: Thank you, we labeled as Figure 1A and 1B.

Text change:

Figure 1A. Two examples of the excluded image: Note the whole length of femur was not included in the image.  

Figure 1B. Fracture of the femur with notable displacement is noted in the image.

Point 5: Experimental Section, line 93: The authors randomly chose 5 male and 5 female radiographs of the 35 available for their study. It is not explained why only 10 of the 35 were studied. Did the authors have sufficient power with n=5 per sex? This should be addressed.

Response 5: We invented a method to measure the curvature of femur. We tested for reliability in ten patients, five men and five women. This study is basically for exposition of the method. We refined our title to make it more explicit and made relevant changes in the whole manuscript to fit our primary goal. We hope our method can be used widely in the future. Large scale multi-center study will be our next step of research.

Text change

(Abstract Section line 29 to 30)

Results: The coefficient of variation of the 240 measurements ranged from 0.007 to 0.295 and averaged 0.088.

(Experimental Section line 167 to 170)

Each measurement was repeated three times by one of the author (C.K.L.) twice with one week interval and then by the other author (Y.P.C.). We calculated the coefficient of variation of each three measurement. Smaller coefficient of variation means good intra-observer and inter-observer reliability.

(Results section line 179 to 183)

The coefficient of variation of these 240 measurements ranged from 0.007 to 0.295 and averaged 0.088. Our analysis showed that the three different measurements on the same radiograph were all highly correlated. The coefficient of variation of 0.088 is rather good for our purpose because the magnification of radiographs are 1.10X to 1.20X, larger than our results.

(Results section line 289)

This study was designed to provide reliability, so the patient number is small.

Point 6: In addition, it should be justified why only 10 of the 35 available samples were selected for study. How does this sample size compare to other previously published studies of femoral curvature?

Response 6: There were 35 femur radiographs meeting the criteria. We randomly selected ten radiographs by computer program. We labeled these 35 radiographs from 1 to 35. Then we randomized its arrangement. Finally we choose the first five male radiographs and first five female radiographs. Comparing with other previous reports, our patient number is frankly small. However, our primary goal is exposition of a new method. Thus we measure every patient three times, one of the author twice with one week interval and the other author. We also calculated the coefficient of variation. After this method is justified, we will collect data in a larger scale study.

Text change

(Abstract Section line 29 to 30)

Results: The coefficient of variation of the 240 measurements ranged from 0.007 to 0.295 and averaged 0.088.

(Experimental Section line 167 to 170)

Each measurement was repeated three times by one of the author (C.K.L.) twice with one week interval and then by the other author (Y.P.C.). We calculated the coefficient of variation of each three measurement. Smaller coefficient of variation means good intra-observer and inter-observer reliability.

(Results section line 179 to 183)

The coefficient of variation of these 240 measurements ranged from 0.007 to 0.295 and averaged 0.088. Our analysis showed that the three different measurements on the same radiograph were all highly correlated. The coefficient of variation of 0.088 is rather good for our purpose because the magnification of radiographs are 1.10X to 1.20X, larger than our results.

(Results section line 289)

This study was designed to provide reliability, so the patient number is small.

Point 7: Experimental Section, line 96: the authors indicate that they built a software program, which is subsequently described in the following paragraphs. However, it is not indicated HOW this program was built (e.g., programming language, operating system it is compatible with, etc.), nor whether the program has been / will be deposited into a publically available database for use by others. These issues must be noted and addressed.

Response 7: With Windows operation system and Delphi 2010 language, we built the software. We can publish this software in the future.

Text change:

(Experiment section line 108 to 110)

With Windows operation system and Delphi 2010 language, we built a software program to read the image which automatically calculated the radii of curvatures of the femur as output after we draw the border of the femur outline and the medullary canal.

Point 8: Figure 2: Actually, I am assuming this is Figure 2, as its current label is “Fig Error! No text or specified style in document.” This must be corrected.

Response 8: Thank you for reminding us. We have corrected it.

Point 9: Experimental Methods, line 128: the “algorithm of territory method” needs to be more fully described, diagramed, and referenced. It is unclear if the authors created this algorithm, or borrowed it from a previously published study. 

Figure 5: the “pink line” is difficult to discern. It is recommended that the authors change this pink line to something “dashed” or otherwise, so it is easier to see when printed in black and white (as many readers may do).

Response 9: This algorithm is a modification usage of region growing method. To explain it, we changed the name from territory method to region growing territory method. We changed the figure 5 too.

Text change:

(Experimental Section line 144 to 149)

We introduced an algorithm of region growing territory method to find the center of the femur outline and the interior medullary canal. The region growing algorithm has been widely used in image segmentation [5, 14, 20, 23]. The central line of the two curves is similar to segmentation. Thus, we modified it to region growing territory method. We set two curves as two seeds. The seeds grow and then meet in the central line. The region growing territory method is described below:

(Figure 5)

Figure 5. Region growing territory method. Solid line 1 represents one curve. Dot line 2 represents another curve. 1 and 2 are seen as seed and grow to two territories (3 and 4). The two territories meet and form central curve. If they do not meet, they will keep on growing.

Point 10: Results, line 154: The authors indicate that all measurements from the same radiograph were highly correlated but do not present data to support this statement. Please provide.

Response 10: We calculated the coefficient of variation and presented in results section.

Text change

(Abstract Section line 29 to 30)

Results: The coefficient of variation of the 240 measurements ranged from 0.007 to 0.295 and averaged 0.088.

(Experimental Section line 167 to 170)

Each measurement was repeated three times by one of the author (C.K.L.) twice with one week interval and then by the other author (Y.P.C.). We calculated the coefficient of variation of each three measurement. Smaller coefficient of variation means good intra-observer and inter-observer reliability.

(Results section line 179 to 183)

The coefficient of variation of these 240 measurements ranged from 0.007 to 0.295 and averaged 0.088. Our analysis showed that the three different measurements on the same radiograph were all highly correlated. The coefficient of variation of 0.088 is rather good for our purpose because the magnification of radiographs are 1.10X to 1.20X, larger than our results.

Point 11: Results, general: At present, the findings from the authors’ study are limited. It may increase the merit of the study to break the results into male and female, then statistically compare the femoral curvature of males vs. females. This would add rigor and interest to the study, rather than simply demonstrating a new tool for measuring femoral curvature, because at present it is unclear from the present results whether this new tool for measuring femoral curvature represents an advance or better alternative to currently available methods as previously published.

Response 11: We have compared the femoral curvature of males and females.

In addition, we compared radii between both genders and observed that the whole length of women’s femur was straighter than men’s (1435mm vs. 1201mm, p=0.002). However, there was no significant difference in the proximal, middle, and distal parts between men and women (p=0.283, 0.456 and 0.135, respectively). The data are shown in Tables 1 and 2.

Table 1. General data (F: female. M: male). Units were mm.

Age

Gender

Whole Outer

Whole Inner

Proximal Outer

Proximal Inner

Middle Outer

Middle Inner

Distal Outer

Distal Inner

24

F

1631

1883

340

811

1883

1404

653

671

27

F

1338

1600

379

1043

1546

1577

983

712

38

F

1105

982

311

632

1089

967

673

452

20

F

1730

1604

432

574

1671

1520

1041

693

31

F

1299

1183

770

1005

1193

1217

604

629

Female’s mean

1421

1450

446

813

1476

1337

791

631

Female’s SD

255

362

186

212

331

248

204

105

37

M

1696

1614

392

632

1486

1812

907

871

40

M

1087

899

436

696

1179

1172

746

1032

43

M

1094

1117

379

795

1418

1166

1098

905

57

M

1112

1038

534

758

1068

1117

715

693

27

M

1330

1019

550

510

1413

1676

558

405

Male’s mean

1264

1137

458

678

1313

1389

805

781

Male’s SD

262

278

80

113

180

329

206

243

Table 2.  Radii of different parts in different groups.

Measurement of femur

Total

(n=60)

Femur   Outline

(n=30)

Medullary Canal (n=30)

Men

(n=30)

Women

(n=30)

P-value

Whole

1318±297

1342±252

1294±338

1201±257

1435±291

0.002

Proximal

752±212

798±210

706±208

793±227

711±192

0.283

Middle

1379±288

1394±285

1363±294

1351±277

1407±300

0.456

Distal

599±220

452±133

746±190

568±161

630±266

0.135

Unit : mm, expressed as mean±SD.

Reviewer 2 Report

Dear Author!

The manuscript "Computerized Measurement of the Sagittal Bowing of Femur from Plain Radiograph in Taiwan" is well designed research providing new method of Sagittal Bowing of Femur meassurement.

The main mistake in my opinion is you conclusion about race difference. So small group for this conclusion. If you modify article focused on new methodology - it will much better!

Below some specific remarks:

Title

Article is dedicated to new method of Sagittal bowing measurement. You did not show any significant difference  between Asian and other populations due to very small number of observation.

My advice is to change title like “New method of….”

Abstract

No statistical description (M±m) The conclusion is pure. 10 cases are so small to provide conclusion about race difference. Again – the best way is to represent new method is the main conclusion

Introduction

Line 41-43 – please, give any references supported this information. Line 44 – need reference

Experimental section

Fig 1 and fig 2 are not give important information and can be excluded from manuscript Provide description of used statistics

Conclusion

1. Need to moderate modification

Author Response

Response to Reviewer 5 Comments

Point 1: The main mistake in my opinion is you conclusion about race difference. So small group for this conclusion. If you modify article focused on new methodology - it will much better!

Response 1: Thank you for the suggestion. We did a major modification to focus on the new method itself.

Text change

(Abstract Section line 29 to 30)

Results: The coefficient of variation of the 240 measurements ranged from 0.007 to 0.295 and averaged 0.088.

(Experimental Section line 167 to 170)

Each measurement was repeated three times by one of the author (C.K.L.) twice with one week interval and then by the other author (Y.P.C.). We calculated the coefficient of variation of each three measurement. Smaller coefficient of variation means good intra-observer and inter-observer reliability.

(Results section line 179 to 183)

The coefficient of variation of these 240 measurements ranged from 0.007 to 0.295 and averaged 0.088. Our analysis showed that the three different measurements on the same radiograph were all highly correlated. The coefficient of variation of 0.088 is rather good for our purpose because the magnification of radiographs are 1.10X to 1.20X, larger than our results.

(Results section line 289)

This study was designed to provide reliability, so the patient number is small.

Point 2: Article is dedicated to new method of Sagittal bowing measurement. You did not show any significant difference between Asian and other populations due to very small number of observation.

My advice is to change title like “New method of….”

Response 2: Thank you. We changed it.

Text change:

(Title)

New Computerized Method in Measuring the Sagittal Bowing of Femur from Plain Radiograph – A Validation Study

Point 3: No statistical description (M±m) The conclusion is pure. 10 cases are so small to provide conclusion about race difference. Again – the best way is to represent new method is the main conclusion

Response 3: We totally agree. We did a major revision. We focused on the new method, not the race difference.

Text change

(Abstract Section line 29 to 30)

Results: The coefficient of variation of the 240 measurements ranged from 0.007 to 0.295 and averaged 0.088.

(Experimental Section line 167 to 170)

Each measurement was repeated three times by one of the author (C.K.L.) twice with one week interval and then by the other author (Y.P.C.). We calculated the coefficient of variation of each three measurement. Smaller coefficient of variation means good intra-observer and inter-observer reliability.

(Results section line 179 to 183)

The coefficient of variation of these 240 measurements ranged from 0.007 to 0.295 and averaged 0.088. Our analysis showed that the three different measurements on the same radiograph were all highly correlated. The coefficient of variation of 0.088 is rather good for our purpose because the magnification of radiographs are 1.10X to 1.20X, larger than our results.

(Results section line 289)

This study was designed to provide reliability, so the patient number is small.

Point 4: Line 41-43 – please, give any references supported this information. Line 44 – need reference

Response 4: We change the paragraph.

Text change:

(introduction section line 42 43)

The difference of sagittal bowing of femur among different individuals or even different ethnicities is important in the present orthopedic practice.

Point 5: Fig 1 and fig 2 are not give important information and can be excluded from manuscript Provide description of used statistics.

Response 5: We show a new computer method in Orthopedics. We think some software engineers are the potential readers and may be interested. Fig. 1 and Fig. 2 will be helpful for engineers.

Round 2

Reviewer 1 Report

The authors have adequately addressed my concerns from the first review. I have no further suggestions for improvement.

Reviewer 2 Report

Dear Authors!

Thank you for detailed revision of manuscript.

In current version it should be published as is